# Parkinson’s Disease and Diabetes Mellitus: Individual and Combined Effects on Motor, Cognitive, and Psychosocial Functions

**DOI:** 10.3390/healthcare11091316

**Published:** 2023-05-04

**Authors:** Jolie D. Barter, Dwaina Thomas, Liang Ni, Allison A. Bay, Theodore M. Johnson, Todd Prusin, Madeleine E. Hackney

**Affiliations:** 1Division of Geriatrics and Gerontology, Department of Medicine, School of Medicine, Emory University, Atlanta, GA 30322, USA; 2School of Arts and Sciences, Clark Atlanta University, Atlanta, GA 30314, USA; 3Department of Family and Preventive Medicine, School of Medicine, Emory University, Atlanta, GA 30322, USA; 4Birmingham/Atlanta VA Geriatric Research Education and Clinical Center, Brookhaven, GA 30319, USA; 5School of Nursing, Emory University, Atlanta, GA 30322, USA; 6Atlanta VA Center for Visual and Neurocognitive Rehabilitation, Decatur, GA 30033, USA; 7Department of Rehabilitation Medicine, School of Medicine, Emory University, Atlanta, GA 30322, USA

**Keywords:** Parkinson’s disease, diabetes, multimorbidity, mobility, executive function

## Abstract

Background/objective: Understanding the effects of multimorbidity on motor and cognitive function is important for tailoring therapies. Individuals with diabetes mellitus (DM) have a greater risk of developing Parkinson’s disease (PD). This study investigated if individuals with comorbid PD and DM experienced poorer functional ability compared to individuals with only PD or DM. Methods: A cross-sectional analysis of 424 individuals: healthy older adults (HOA), n = 170; PD without DM (PD-only), n = 162; DM without PD (DM-only), n = 56; and comorbid PD and DM (PD+DM), n = 36. Motor, motor–cognitive, cognitive, and psychosocial functions and PD motor symptoms were compared among groups using a two-way analyses of covariance with PD and DM as factors. Results: The PD-only and DM-only participants exhibited slower gait, worse balance, reduced strength, and less endurance. Motor–cognitive function was impaired in individuals with PD but not DM. DM-only participants exhibited impaired inhibition. Individuals with comorbid PD+DM had worse PD motor symptoms and exhibited impaired attention compared to the PD-only group. Conclusions: Having PD or DM was independently associated with poorer physical and mental quality of life, depression, and greater risk for loss of function. Both PD and DM have independent adverse effects on motor function. Comorbid PD+DM further impairs attention compared to the effect of PD-only, suggesting the importance of therapies focusing on attention. Understanding the functional ability levels for motor and cognitive domains will enhance the clinical care for PD, DM, and PD+DM.

## 1. Introduction

Similar molecular pathways (e.g., inflammation, mitochondrial function, and metabolism) are dysregulated in both Parkinson’s disease (PD) and diabetes mellitus (DM) [1]. Individuals with type 2 DM may have an increased risk of developing PD and likely share a similar molecular mechanism [2]. The available evidence is, in fact, conflicting [3,4]. One study was able to demonstrate that pre-existing type 2 diabetes contributes to faster disease progression and reduced survival in PD [5]. PD and DM can similarly negatively affect motor, cognitive, and psychosocial function [6]. Individuals with PD experience impaired motor (bradykinesia, rigidity, tremors, postural instability, and gait impairment), cognitive (spatial cognition and executive function), and motor–cognitive (ability to integrate and modulate complex cognitive and motor skills) functions [7,8]. Similar to PD, DM can lead to decreased mobility [9], and there is a 73% increased risk of dementia for individuals with DM [10]. Initial work has indicated that DM impairs dual tasking [11]. PD and DM manifesting in the same individual could exacerbate the overall negative effects of each condition [12] on a host of health factors, including rehabilitation-relevant motor, cognitive, and psychosocial functions.

Previous work has, indeed, shown that people with comorbid PD and DM (PD+DM) exhibit higher Unified Parkinson’s Disease Rating Scale (UPDRS) motor scores compared to PD only [13]. Comorbid PD+DM can impair global cognition compared to PD only [14,15]. However, the cognitive domains specifically affected are disputed. One study reported impaired executive function [14], but another study found no difference [15].

This study evaluated clinical measures of motor, motor–cognitive, cognitive, and psychosocial functions in 424 individuals, including healthy older adults (HOA), n = 170; PD without DM (PD-only), n = 162; DM without PD (DM-only), n = 56; and comorbid PD and DM (PD+DM), n = 36. While previous research demonstrates that comorbid PD+DM leads to more severe PD motor symptoms, this study evaluated the effect of comorbid PD+DM on clinical measures of motor function (endurance, strength, and dynamic balance) and motor–cognitive integration, which has yet to be assessed. Due to the inconsistent previous reports, this study also assessed which cognitive domains (attention, planning, spatial, and organization) are more affected in individuals with comorbid PD+DM. The measures of PD severity were also assessed in individuals with PD only and comorbid PD+DM to corroborate previous research. Unlike previous work, the control DM-only and the HOA groups were included to strengthen the rigor of our approach. By understanding how these clinical measures change with comorbid PD+DM, we can assess health status (i.e., fall risk, mobility, functional ability, and risk of loss of function) to better understand how to enhance the clinical care for individuals with comorbid PD+DM.

## 2. Methods

### 2.1. Study

This study used retrospective secondary data collected from observational and rehabilitative studies conducted from 2011 to 2019. The studies were approved by the Institutional Review Board at Emory University School of Medicine and the Research and Development Committee of the Atlanta Veterans Affairs (VA) Medical Center. The IRB numbers for the studies are IRB00097348, IRB00080676, IRB00055977, IRB00060613, and IRB00047231. All participants provided written informed consent. Paper and electronic files were de-identified and coded to maintain confidentiality. The funders played no role in the design, conduct, or reporting of this study.

Data from 540 consenting participants were considered for the trial. Data from 116 were excluded from the final analyses primarily because of a lack of the availability of data from several assessments. Therefore, we present data from 424 participants. The progress through the trial is detailed in the STROBE flow diagram in Figure 1.

### 2.2. Participants

A total of 424 individuals participated, including 170 HOA, 162 PD only, 56 DM only, and 36 comorbid PD+DM (Table 1). Participants were recruited through the Atlanta VA Center for Visual and Neurocognitive Rehabilitation registry, the VA Informatics and Computing Infrastructure database, Michael J. Fox’s Foxfinder website, the Movement Disorders unit of Emory University, PD organizations’ newsletters, support groups, educational events, and word of mouth. All eligible participants were aged 40–85 years, could walk at least 3 m with or without assistance, had visual acuity of 20/40 or better in the best eye, and were cognitively able to provide their own consent to the study. In addition to these criteria stated above, PD participants must also have been clinically diagnosed with PD by a board-certified neurologist with specialty training in movement disorders, exhibited 3 of the 4 cardinal signs of PD, exhibited clear symptomatic benefit from antiparkinsonian medications, and in Hoehn and Yahr stages I–IV. The participants with PD were tested in either the ON or OFF states, i.e., at least 12 h after their last antiparkinsonian medications. Of the 198 individuals with PD in this study, there were 84 participants tested while OFF their medication and 114 individuals tested while ON their medication. DM status was determined through self-report on a health questionnaire. Chronic conditions that were reported by participants, including DM, were verified as being treated by the participants’ medical providers.

### 2.3. Measures Overview

Measures of PD severity, motor function, mobility, motor–cognitive function, cognitive function, and psychosocial function were assessed during one session (lasting 2–3 h depending on the needs and timing of the participant) in invariant order. Participants were allowed to take breaks ad libitum. Hundreds of participants with DM and PD have been seen by this lab in similar batteries, which has been determined to be tolerable. Clinically, mobility is strictly defined as the ability of a joint to move freely through a given range of motion without restriction from surrounding tissues [16]. This study measured mobility in multiple ways using standard clinical physical therapy assessments of gait speed and dynamic balance. Sociodemographic factors (age, education, number of falls, comorbidities, medications, years with PD, race, housing, transportation frequency of leaving the house, freezer status, and use of assistive device) were collected via a project health questionnaire [17,18]. Fear of falling and QOL were measured with a 7-point scale, from 1 (low) to 7 (high) [19]. The assessments were performed the same as previously published methods [20].

### 2.4. PD Severity

The PD severity was measured using the UPDRS part III or the MDS-UPDRS part III (motor exam) [21], Parkinson’s Disease Questionnaire (PDQ-39) [22], Freezing of Gate Questionnaire (FOGQ) abbreviated [23], and time with PD (years) as reported in the project health questionnaire. We used standard, published methods of conversion to convert the UPDRS scores (when available) into MDS-UPDRS scores [24].

### 2.5. Motor Function

The average preferred forward (forward), fast forward (fast), and preferred backward (backward) gait speeds (meters/sec, m/s) were derived from 3 trials of walking over 20 feet [25,26]. The participants were also assessed using the Fullerton Advanced Balance Scale and the Berg Balance Scale, including the 30 s chair stand test [27], the 6 min walk test (6MWT) [17], the 360° turn test, and the one leg stand test [28].

### 2.6. Motor–Cognitive Function

Dual-task timed up and go (TUG) is a valid measure of functional dual-task abilities [29]. Individuals are asked to rise from a chair, walk 3 m to cross a line, turn around, and return to the chair as quickly and safely as possible (TUG-baseline or single task). Participants were timed while they performed a dual task: TUG-baseline and a concurrent cognitive component (TUG-cognitive or dual-tasking: counting backward by 3 from a random number between 20 and 100) [30]. The amount of time to complete the task and the number of correct subtractions per second were used for analysis (correct subtractions per second). The four square step test (FSST) assessed movement planning, coordination, and speed [31]. Rods were arranged into a cross to create four squares. The participants stepped over the rods in a clockwise then counterclockwise manner. The time it took to complete the task was recorded. The body position spatial test (BPST) is a whole-body spatial cognition test involving multidirectional steps and turns in lengthening sequences [20]. The examiner verbally and visually demonstrates a combination of side and forward steps and turns in place. The task ends once a participant misses both trials in a level. The product (product score) of the length of moves correctly performed (span) and number of trials performed correctly (trials) were used for analysis.

### 2.7. Cognitive Function

The serial 3 subtraction test assessed mental attention [32]. Other assessments included the Montreal cognitive assessment (MoCA), trail making tests (parts A and B) (difference score used in the analyses) [33], Delis–Kaplan executive function system (D-KEFS), color–word interference test (CWIT), and tower test [34]. Raw scores were converted to scaled scores that were normalized for age. For the CWIT analysis, the scaled scores from each of the four conditions (color naming, word reading, inhibition, and inhibition/switching) were calculated. For the tower test analysis, the total achievement score, time per move ratio scaled, and mean time to make the first move were calculated. The Brooks spatial memory task measured spatial cognition using mental imagery [35]. The reverse Corsi blocks task measured visuospatial working memory [36]. Like BPST, a product score was calculated.

### 2.8. Psychosocial Function

The psychosocial measures administered were the Short-Form Health Survey–12 (SF-12) (/100, higher = better health) using the physical (PCS) and mental composite (MCS) scales [37]; life space questionnaire (LSQ) (/9, higher = lower mobility) [38]; physical activity scale for the elderly (PASE) (/400, higher = more physical activity) [39]; composite physical function (CPF) questionnaire (/24, higher = less able to perform ADLs); Beck depression inventory–II (BDI-II) (/63, higher = more symptoms of depression) [40]; and activities-specific balance confidence scale (ABC) (/100; higher = confident they will not lose their balance) [41].

The composite physical function (CPF) questionnaire is a health questionnaire that characterizes participants’ general ability to complete activities of daily living (ADLs) with the composite physical function index (CPF) (/24; higher = better), and it was used to ascertain participants’ ability to perform ADLS and their general physical health.

### 2.9. Statistics

The variables were tested for normal distribution with histograms and Kolmogorov–Smirnov tests. For the demographics, one-way analyses of variance (ANOVAs) and t-tests compared four groups and two groups, respectively. For group comparison of categorical variables, chi-square tests were used. The sample had differences among groups in sociodemographic factors: education, sex, and race. A significant difference was also observed for time with PD. Therefore, we controlled for these factors (i.e., sex, education, and race) in comparisons of the four groups on continuous outcome variables using a stepwise linear regression with two-way analyses of covariance (ANCOVAs). Four models were produced covarying (controlling) for confounders: model 1—education; model 2—education and sex; model 3—education, sex, and race; and model 4—education, sex, race, and time with PD. Adjusted *p*-values were examined for main effects and interactions for the factors “Parkinson’s disease” and “diabetes”. Only *p*-values from model 4 are presented. All *p*-values can be found in the Appendix A. Depending on the normality, unadjusted *p*-values were obtained using a two-factor analysis of variance (ANOVA) or an aligned rank transformation two-factor ANOVA. Post hoc analyses were performed using Tukey’s test. All data were used for these analyses. Some observations were not available because the same assessments were not administered in all studies, or the participant refused or were unable to perform the assessment. Tests were two-tailed with alpha = 0.05. A supplemental analysis was performed using independent t-tests to determine whether performance on motor and cognitive assessments differed among Parkinson’s individuals either ON PD medication or OFF PD medication (12 h since the last dose). Psychosocial assessments were not included in this analysis because all participants were ON medications during these assessments. The analyses were performed using R version 3.6.1. To avoid ambiguity of the terms with respect to the effect versus group, we used the following conventions: “HOA”, “PD-only”, “DM-only”, and “PD+DM” when referring to groups and “Parkinson’s” and “diabetes”, and “PDXDM” when referring to main effects. The interpretation intervals for the eta squared are consistent with the following conventions: *η*^2^ = 0.01 indicates a small effect; *η*^2^ = 0.06 indicates a medium effect; and *η*^2^ = 0.14 indicates a large effect.

## 3. Results

A total of 424 participants met the eligibility criteria and were available for the assessments: 170 HOA, 162 PD-only, 56 DM-only, and 36 PD+DM (Table 1). A significant difference among groups was found for years of education (*p* = 0.002), BMI (*p* < 0.001), number of medications (*p* < 0.001), number of comorbidities (*p* < 0.001), sex (*p* < 0.001), and race (*p* < 0.001). The PD-only participants had more years of education compared to the DM-only (*p* < 0.001) and HOA (*p* = 0.02) groups. Higher BMIs were observed for HOA (*p* < 0.001), DM-only (*p* = 0.01), and PD+DM (*p* < 0.001) compared to PD-only. HOA took fewer medications compared to the other groups (*p* < 0.001). PD+DM had more comorbidities compared to PD-only (*p* < 0.001) and HOA (*p* < 0.001). More women were in the HOA and DM-only groups. More men and White individuals were in the PD groups.

Concerning only the PD groups, PD+DM had higher UPDRS-III scores (*p* = 0.039) and fewer years with PD (*p* = 0.023) than PD-only (Table 1).

### 3.1. Motor Assessments

Effect of Parkinson’s disease: The main effects of Parkinson’s were observed for the gait speed, chair stand, 6MWT, and 360° turn tests (Table 2). People with PD (including individuals from the PD-only and PD+DM groups) had slower fast (*p* = 0.011, *η*^2^ = 0.017) and backward (*p* = 0.003, *η*^2^ = 0.022) gait speeds. People with PD stood up from a chair fewer times (*p* = 0.024, *η*^2^ = 0.013), walked less far in 6 min (*p* = 0.009, *η*^2^ = 0.018), and took more steps (*p =* 0.002, *η*^2^ = 0.037) and more time (*p* = 0.026, *η*^2^ = 0.021) to turn 360° (Figure 2 and Figure 3).

Effect of diabetes: The main effects of diabetes were observed for the fast gait speed, chair stands, 6MWT, and one leg stand tests (Table 2). People with DM (including individuals from the DM-only and PD+DM groups) had slower fast gait speed (*p* = 0.042, *η*^2^ = 0.011), performed fewer chair stands (*p* = 0.049, *η*^2^ = 0.01), walked less far in 6 min (*p* = 0.001, *η*^2^ = 0.026), and stood for less time on one leg (*p* = 0.023, *η*^2^ = 0.018) (Figure 2 and Figure 3).

### 3.2. Motor–Cognitive Assessments

Effect of Parkinson’s disease: The main effects of Parkinson’s were observed for TUG-baseline and TUG-cognitive (Table 3). People with PD completed the TUG-baseline more slowly (*p* = 0.007, *η*^2^ = 0.017) and during the TUG-cognitive test made fewer correct subtractions per second (*p* = 0.004, *η*^2^ = 0.021) (Figure 3).

Effect of diabetes: A main effect of diabetes was observed for the TUG-baseline task (Table 3). People with DM completed the TUG-baseline more slowly (*p* = 0.04, *η*^2^ = 0.01). (Figure 3).

### 3.3. Cognitive Assessments

Interaction effect between Parkinson’s disease and diabetes (PDXDM): A significant interaction between PD and DM was observed for the serial 3s task (*p* = 0.033, *η*^2^ = 0.01), CWIT inhibition scaled score (*p* = 0.038, *η*^2^ = 0.015), tower test achievement score (*p* = 0.036, *η*^2^ = 0.015), and time per move (*p* = 0.001, *η*^2^ = 0.037) (Table 3). The post hoc analyses indicated a higher percent correct for the serial threes test for PD-only compared to PD+DM (*p* = 0.02). A poorer performance on the CWIT inhibition task was observed in DM-only (*p* = 0.006) compared to HOA. The time per move was longer for PD-only (*p* = 0.042) and DM-only (*p* = 0.047) compared to PD+DM on the tower test (Figure 4).

Effect of Parkinson’s disease: The main effects of Parkinson’s were observed for the tower test achievement score and the tower test mean first move time (Table 3). People with PD had higher achievement scores (*p* = 0.033, *η*^2^ = 0.015) and took less time to make the first move (*p* = 0.003, *η*^2^ = 0.029).

Effect of diabetes: The main effects of diabetes were observed for the CWIT inhibition scaled score and the tower test mean first move time (Table 3). People with DM exhibited impaired inhibition (*p* = 0.01, *η*^2^ = 0.024) and took longer to make the first move for the tower test (*p* = 0.024, *η*^2^ = 0.017).

### 3.4. Psychosocial Assessments

Effect of Parkinson’s disease: The main effects of Parkinson’s were observed for the SF-12, CPF, BDI-II, and fear of falling tests (Table 4). People with PD had lower SF-12 PCS (*p* < 0.001, *η*^2^ = 0.027) and CPF (*p* = 0.049, *η*^2^ = 0.008) scores, indicating poorer physical QOL and functional ability. People with PD had higher BDI-II (*p* = 0.013, *η*^2^ = 0.016) scores, indicating lower mental QOL and some depression (Figure 4).

Effect of Diabetes: The main effects of diabetes were observed for the SF-12 PCS, CPF, and BDI-II tests (Table 4). People with a diagnosis of DM had lower SF-12 PCS (*p* = 0.004, *η*^2^ = 0.018) and CPF (*p* = 0.005, *η*^2^ = 0.017) scores, indicating poorer physical QOL and functional ability. People with DM had higher BDI-II (*p* = 0.018, *η*^2^ = 0.014) scores, indicating possible depression (Figure 4).

### 3.5. Clinical Outcome Measures: Falls

Effect of Parkinson’s disease: The main effect of Parkinson’s was observed for the fear of falling test (Table 4). Individuals with PD had greater fear of falling (*p* = 0.012, *η*^2^ = 0.013). While not statistically significant, the PD-only group fell on average 11 times per year compared to the other groups, which averaged approximately 2–3 falls per year.

### 3.6. Supplemental Analysis of Medication Status

The supplemental analysis evaluating the differences in performance based on Parkinson’s medication states found that individuals who were tested while ON their Parkinsonian medication had a faster gait backward speed, took fewer steps and less time to turn 360 degrees, had higher Corsi product scores, and made more correct subtractions per second during the TUG-cognitive test compared to individuals who were tested while OFF their Parkinsonian medication (Appendix A).

## 4. Discussion

In this sample, Parkinson’s and diabetes had independent adverse effects on motor, cognitive, and psychosocial functions and apparently minimal additive adverse effects on cognition and PD motor symptoms. Significant interactions showed two areas that did appear to be affected by combined PD and DM: attention and disease severity; although the PD+DM group had fewer years with PD than the PD-only group, and participants with comorbid PD+DM exhibited more severe Parkinsonian motor symptoms and reduced attention compared to PD-only participants. A significant interaction also showed that people with DM had worse inhibition compared to healthy older adults without DM. This work is important because (A) our findings corroborate previous findings, which has important scientific and clinical merit in its own right; (B) we strengthen and highlight information that suggests that many people with DM suffer from motor impairments that warrant attention clinically from physiatrists and physical therapists—given that those with DM had a motor performance that was not unlike those with PD; and (C) we present motor and cognitive performance on several standard, reliable, and validated neuropsychological measures that have not previously been presented in investigations concerning a comparison of these populations.

Consistent with previous work [20], this study observed that individuals with PD had slower gait speed, shorter endurance, reduced leg strength, and poorer balance. The most compelling results regard the DM group, which is not conventionally deemed a motor disorder. These results in people with DM show impaired gait speed, balance, and endurance. These findings add to a growing body of research showing similar traits [9,42] and suggests that DM should be thought of as a movement disorder. Impaired motor function could result from peripheral neuropathy [43], which affects ~50% of individuals with DM [44] and ~16.3% of individuals with PD [45]. Here, self-reported peripheral neuropathy was very low (1/162 PD-only and 1/56 DM-only). Individuals with DM exhibited impaired inhibition and slower organization/planning skills. DM is associated with increased risk of developing cognitive impairments [10]. While previous studies report a decline in executive function with DM [46]. This paper is the first to report that older diabetics exhibited impaired performance on the DKEFS CWIT inhibition test and slower first move time for the DKEFS tower test.

As with previous studies [8], here, individuals with PD exhibited impaired motor–cognition per the TUG-cognitive test. Previous literature has shown that in the PD populations, attention is important for successful dual-tasking [47]. No significant effect of diabetes was observed on motor–cognitive integration, although other studies have shown that diabetics with peripheral neuropathy exhibited impaired gait in a dual-task compared to diabetics without neuropathy [11].

The results show that individuals with PD had a greater fear of falling. The PD-only group fell approximately 11 times per year compared to HOA (<1 fall per year), possibly due to the fact of poorer motor–cognitive integration [47]. Prior work showed that diabetics fall more compared to older adults [9]. This study reports that people with DM fell on average three times per year compared to HOA (<1 fall per year). A strong correlation was observed between the fear of falling and lower QOL (R^2^ = 0.083) [48], indicating the importance of reducing fear of falling in all four of the groups examined in this study.

Individuals with either a PD or DM diagnosis reported performing less physical activity, performing fewer ADLs, and exhibiting depression. Prior work has shown that 35% of people with PD [49] and 29% of people with DM [50] exhibit depressive symptoms. Lower psychosocial function with PD or DM is consistent with previous literature [51,52]. Our findings add to the urgency to develop better diagnostic methods and treatment methods for people dealing with comorbid PD and DM, as well as mood disorders.

More severe PD motor symptoms were observed in the comorbid PD+DM group, although PD+DM had fewer years with PD than those with PD-only. In addition to another report showing increased disease severity in people with PD+DM [13], individuals with comorbid PD+DM had higher scores on the UPDRS part III compared to PD-only. This poorer motor function should be addressed clinically by physicians when treating people with both PD and DM. Another report showed that the progression of PD motor symptoms could become more severe sooner with comorbid PD+DM [53]. Moreover, comorbid PD+DM exhibited impaired cognition. Specifically, the PD+DM group exhibited greater impaired attention (as per performance on the serial 3s test) compared to the PD-only group. This study adds to the concerning evidence already amassed that comorbid PD+DM causes faster progression and more severe PD motor symptoms [5]. In addition, the present study examined and reports for the first time specific cognitive domains (attention) that are impaired in comorbid PD+DM compared to PD-only.

## 5. Limitations

While there were differences among the groups observed for education, sex, and race, we used these factors as careful covariates in our analyses to address all confounds. In addition, observing more males in the PD groups is appropriate because of the epidemiological prevalence in the PD population, which is 60 male:40 female [54]. While slightly more diabetics were women, DM is typically more prevalent in males, and there is little difference in sex with DM diagnosis by the 7th decade [55]. Future studies should collect information on the duration of diabetes at the time of the study, efficacy of glycemic control in the enrolled participants, exposure to rehabilitation treatment in the previous months, and the number of diabetes-related complications. Another limitation is the cross-sectional analysis design. The results suggest that the severity of PD motor symptoms develop quicker with a co-diagnosis of DM. A longitudinal study would be more appropriate to evaluate the progression of these conditions. We do not know if DM predated PD diagnosis, as well as the type of diabetes. However, previous studies have demonstrated that in the older adult population, overwhelmingly, type 2 diabetes is the most prevalent type [56]. Another limitation is the possible effect that dopaminergic and hypoglycemic drugs could have on performance of motor and cognitive function [57].

These participants had been under the medical care of physicians for years; therefore, the cross-sectional nature of these data is impacted by the patients’ medical treatments in both known and unknown ways. This possibility represents one challenge of conducting research in chronically ill participants with multiple morbidities, a key focus of current and future geriatric research. While the differences between ON and OFF states were few (5/27 analyses were significant: individuals ON Parkinsonian medication had faster backward gait speed, needed fewer steps and less time to turn 360 degrees, had higher Corsi product scores, and made more correct subtractions per second during the TUG-cognitive test compared to individuals who were tested while OFF their Parkinsonian medication (Appendix A)), the limitation remains that in individuals with PD who are OFF medication, this could be causing the main effects of PD, and motor impairment may have been aggravated in an OFF state.

## 6. Conclusions

Even though DM is not thought of as a motor disease, our data add to evidence demonstrating that people with DM exhibit decreased motor function. Our data in these participants with comorbid PD+DM suggest the dual conditions may lead to more severe PD motor symptoms and poorer attention function, but more research is certainly warranted. In fact, measures of clinical mobility (gait, strength, and dynamic balance) were not further impaired in individuals with comorbid PD+DM compared to PD-only, which is a surprising finding. The negative effect of comorbid PD+DM on cognitive function indicates the importance of employing cognitive therapies to improve clinical care for individuals with comorbid PD+DM.

## Figures and Tables

**Figure 1 healthcare-11-01316-f001:**
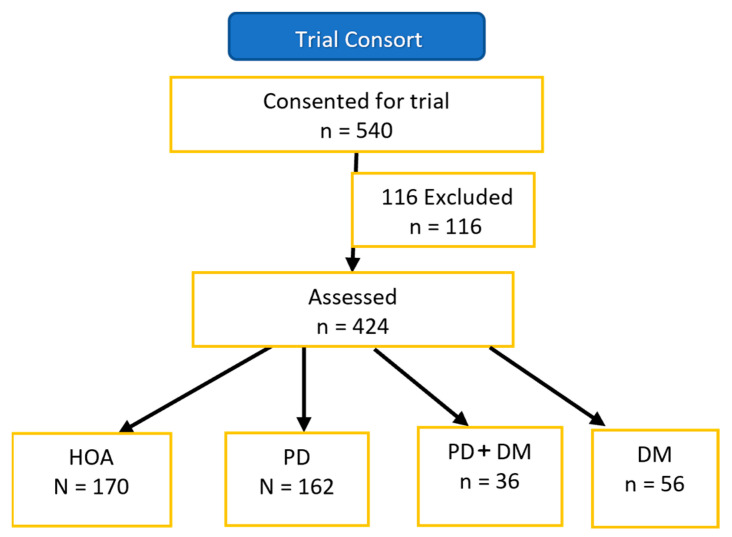
STROBE flow diagram for the trial. Abbreviations: PD+DM = Parkinson’s Disease and Diabetes Mellitus; HOA = Healthy Older Adults; PD-only = Parkinson’s Disease only; DM = Diabetes Mellitus.

**Figure 2 healthcare-11-01316-f002:**
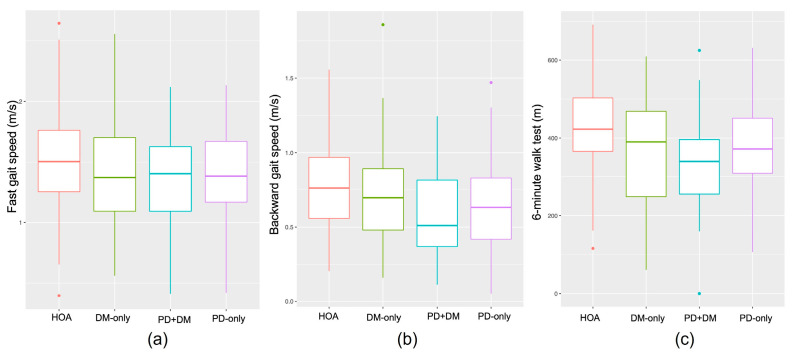
Mobility performance for Healthy Older Adults (HOA) (red), Diabetes Mellitus (DM)-only (green), PD+DM (blue), and PD-only (purple) on fast as possible gait speed (**a**), backward gait speed (**b**) and 6 min walk test (**c**).

**Figure 3 healthcare-11-01316-f003:**
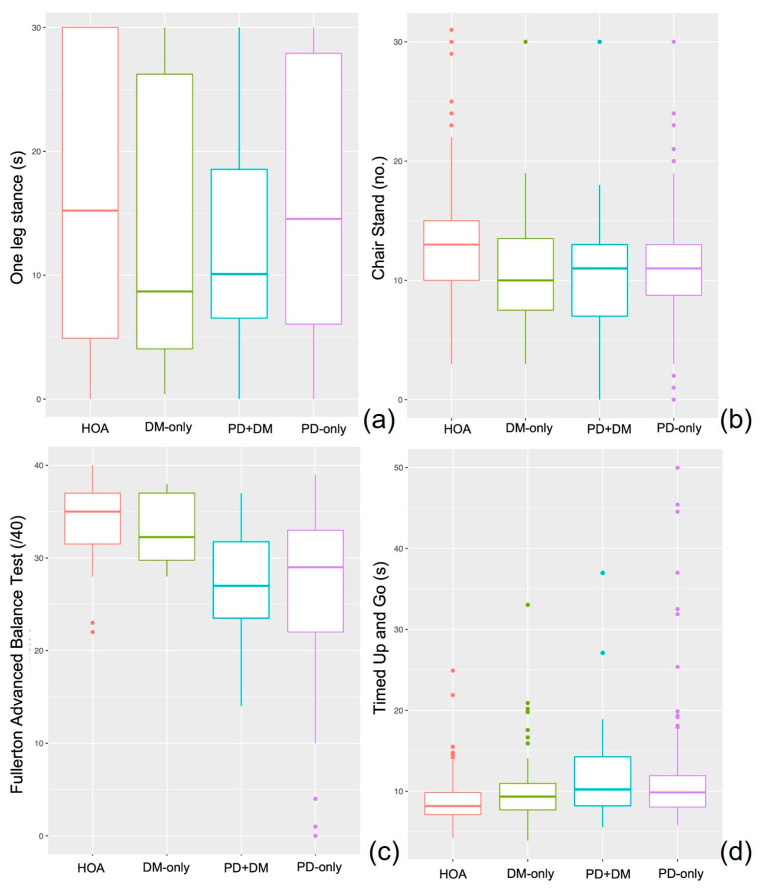
Balance and Mobility performance for HOA (red), DM-only (green), PD+DM (blue), and PD-only (purple) on one leg stance (**a**), 30 s chair stand (**b**), Fullerton advanced balance scale (**c**) and Timed Up and Go (**d**).

**Figure 4 healthcare-11-01316-f004:**
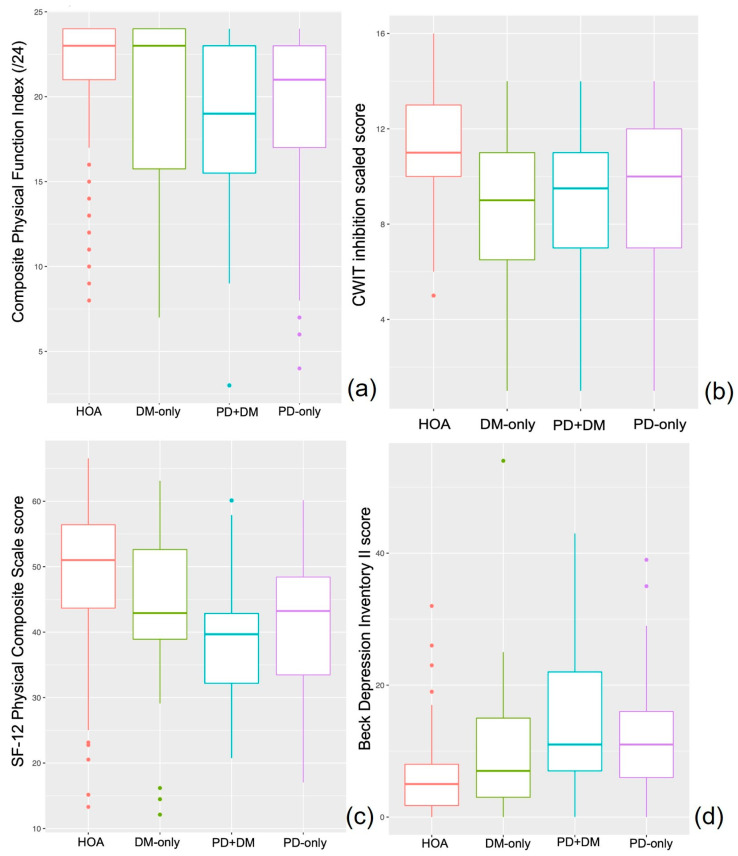
Quality of life and Cognitive performance for HOA (red), DM-only (green), PD+DM (blue), and PD-only (purple) on Composite Physical Function (**a**), Color Word Interference task inhibition scaled score (**b**), Shortform 12 Physical Composite scale (**c**), and Beck Depression Inventory II score (**d**).

**Table 1 healthcare-11-01316-t001:** Participant Demographics and Measures of PD Severity.

		HOA	PD-Only	DM-Only	PD+DM	*p*-Value
n		170	162	56	36	
Age		69.1 ± 9.8	69.49 ± 8.1	68.89 ± 9.1	69.94 ± 7.6	0.927
Education (years)		15.6 ± 2.7	16.42 ± 2.3	15.07 ± 2.4	16 ± 2.7	0.002
Body mass index (kg/m^2^)		27.94 ± 6.1	25.73 ± 4.6	28.59 ± 5.2	29.07 ± 5.3	<0.001
Number of medications		2.78 ± 2.6	5.64 ± 4.1	5.26 ± 3.2	7.54 ± 3.9	<0.001
Number of comorbidities		2.39 ± 1.6	3.11 ± 1.7	5.16 ± 2.8	4.97 ± 1.7	<0.001
Time with PD (years) ^†^			7.09 ± 4.9		5.17 ± 3.8	0.023
Unified Parkinson’s disease Rating Scale III			33.21 ± 11.5		37.79 ± 13.9	0.039
PDQ 39 Summary Index ^†^			21.93 ± 13.8		27.97 ± 17.5	0.075
FOGQ ^†^			6.3 ± 5.1		7.1 ± 5.9	0.678
Sex (n (%))						<0.001
	Men	30 (17.6)	95 (58.6)	22 (40)	30 (83.3)	
	Women	140 (82.4)	67 (41.4)	33 (60)	6 (16.7)	
Race (n (%))						<0.001
	Black	89 (52.7)	25 (15.4)	32 (58.2)	8 (22.2)	
	White	61 (36.1)	128 (79)	20 (36.4)	24 (66.7)	
	Other	19 (11.2)	9 (5.6)	3 (5.5)	4 (11.1)	
Hoehn and Yahr (n (%)) ^						0.888
	1		6 (4.3)		1 (3.6)	
	1.5		18 (12.9)		3 (10.7)	
	2		57 (40.7)		14 (50)	
	2.5		22 (15.7)		3 (10.7)	
	3		35 (25)		6 (21.4)	
	4		2 (1.4)		1 (3.6)	

Values are presented as the mean ± SD for continuous variables and n (%) for categorical variables. ^†^ A nonparametric test was used. FOGQ, freezing of gate questionnaire; MDS, Movement Disorder Society; PDQ, Parkinson’s disease questionnaire. Alpha was set at *p* < 0.05. ^ Data were missing for Hoehn and Yahr for n = 22 for PD-only and n = 8 for PD+DM.

**Table 2 healthcare-11-01316-t002:** The effect of a single morbidity (PD-only and DM-only) and comorbidity (PD+DM) on motor and motor–cognitive functions.

	HOA		PD-Only		DM-Only		PD+DM		*p*-Values Adjusted for Model 3/Model 4		
	n	Mean ± SD	n	Mean ± SD	n	Mean ± SD	n	Mean ± SD	Main Effect of PD	Main Effect of DM	PDXDM
Preferred forward gait speed (m/s)	145	1.09 ± 0.2	140	1.01 ± 0.3	47	0.99 ± 0.3	26	0.97 ± 0.3	**0.001/**	0.062	0.34
Fast forward gait speed (m/s)	145	1.53 ± 0.4	140	1.39 ± 0.4	46	1.39 ± 0.4	26	1.36 ± 0.5	**<0.001**	**0.048**	0.286
Preferred backward gait speed (m/s)	145	0.77 ± 0.3	139	0.63 ± 0.3	46	0.71 ± 0.3	26	0.57 ± 0.3	**<0.001**	0.105	0.878
Chair stands	148	13.01 ± 5	140	10.98 ± 4.9	47	10.94 ± 5.1	27	10.41 ± 5.7	**<0.001**	0.062	0.255
6MWT (m)	144	422.9 ± 105	135	374.7 ± 102	47	366.7 ± 137	27	334.31 ± 132	**<0.001**	**0.002**	0.562
One leg stand (s)	82	16.7 ± 11.5	131	15.97 ± 10.7	28	13.72 ± 11.6	24	11.96 ± 8.2	0.398	**0.03**	0.504
360° turn (steps)	66	3.02 ± 1.2	117	5.85 ± 5.7	18	4.5 ± 5.1	18	4.91 ± 2.7	**<0.001**	0.926	0.11
360° turn (s)	66	6.21 ± 2.2	117	11.26 ± 6.4	17	6.59 ± 2.9	18	10.22 ± 4.1	**<0.001**	0.498	0.257
TUG-baseline (s)	158	8.79 ± 2.7	154	11.43 ± 6.8	49	10.6 ± 5	33	12.13 ± 6.5	**<0.001**	**0.049**	0.422
TUG-cognitive (s)	157	12.79 ± 5.0	153	16.21 ± 15.1 *	49	14.51 ± 6.0	32	15.83 ± 8.3	**0.001**	0.377	0.44
TUG-cognitive (Correct subtractions/s)	157	0.3 ± 0.2	152	0.26 ± 0.3	49	0.26 ± 0.2	32	0.24 ± 0.3	**0.002**	0.218	0.731
FSST (s)	158	11.13 ± 5.1	134	12.15 ± 4.6	41	11.32 ± 3.7	30	12.57 ± 5.2	0.051	0.837	0.874

MWT, minute walk test; m/s, meters per second; TUG, timed up and go; FSST, four square step test; s, seconds. *p* < 0.05 are in bold. * This standard deviation is correct (some individuals who had an episode of freezing of gait needed considerable time to complete the TUG-cognitive).

**Table 3 healthcare-11-01316-t003:** The effect of a single morbidity (PD-only and DM-only) and comorbidity (PD+DM) on cognitive function.

	HOA		PD-Only		DM-Only		PD+DM		*p*-Value Adjusted for Education, Sex, and Ethnicity		
	n	Mean ± SD	n	Mean ± SD	n	Mean ± SD	n	Mean ± SD	Main Effect of PD	Main Effect of DM	PDXDM
Serial 3s counting (% correct)	158	88.7 ± 18.9	155	93.46 ± 11	51	87.12 ± 20.8	34	83.54 ± 25.3	0.558	0.051	**0.035**
MoCA score	159	24.87 ± 3.9	143	25.34 ± 3.8	45	23.69 ± 3.1	34	24 ± 4.7	0.569	0.143	0.995
CWIT inhibition scaled score	98	11 ± 2.4	91	9.08 ± 3.9	27	8.56 ± 3.5	22	8.82 ± 3.7	**0.034**	**0.034**	**0.012**
CWIT inhibition/switching scaled score	97	10.06 ± 3.2	89	8.76 ± 4.1	26	8.54 ± 3.1	21	8.95 ± 3.4	**0.009**	0.486	0.068
Tower test—achievement score	112	10.1 ± 2.5	111	10.05 ± 3.4	32	9.25 ± 3.1	31	11.32 ± 2.5	0.719	0.343	**0.014**
Tower test—time per move (s/total moves)	112	9.4 ± 3.2	111	7.99 ± 4	32	7.66 ± 4.1	31	10.23 ± 4	0.082	0.477	**<0.001**
Tower test—mean first move (s)	112	9.67 ± 2.9	111	10.87 ± 3.5	32	9.94 ± 3.2	31	12.68 ± 2.6	**0.005**	**0.014**	0.077

MoCA, Montreal cognitive assessment; CWIT, color–word interference test; s, seconds. *p* < 0.05 are in bold.

**Table 4 healthcare-11-01316-t004:** The effect of a single morbidity (PD-only and DM-only) and comorbidity (PD+DM) on psychosocial function and clinical outcomes.

	HOA		PD-Only		DM-Only		PD+DM		*p*-Value Adjusted for Education, Sex, and Ethnicity		
	n	Mean ± SD	n	Mean ± SD	n	Mean ± SD	n	Mean ± SD	Main Effect of PD	Main Effect of DM	PDXDM
SF-12 (PCS)	160	49.21 ± 9.7	136	41.52 ± 9.4	50	44.45 ± 11.6	32	38.85 ± 10.1	**<0.001**	**0.006**	0.533
SF-12 (MCS)	160	49.32 ± 8.9	136	45.8 ± 9.9	50	47.69 ± 10.4	32	42.89 ± 10.1	**<0.001**	0.19	0.695
Life space questionnaire	95	6.27 ± 1.1	50	6.66 ± 1.3	33	6.09 ± 1.5	11	5.91 ± 1	0.641	0.229	0.223
PASE	124	108.5 ± 68.9	136	105.7 ± 71.1	33	111.1 ± 78.7	34	89.31 ± 67.2	0.29	0.235	0.275
CPF	165	21.27 ± 4.1	159	19.19 ± 5	56	19.39 ± 5.5	35	18.37 ± 5.4	**<0.001**	**0.010**	0.361
BDI-II	124	5.99 ± 5.7	136	12.32 ± 7.7	33	10.03 ± 10.6	33	14.21 ± 10.9	**<0.001**	**0.038**	0.273
ABC	126	60.67 ± 28.4	159	68.85 ± 24.1	39	58.47 ± 31.3	35	59.85 ± 26.4	0.391	0.108	0.388
Quality of life ^§^	161	5.51 ± 1.22	157	5.10 ± 1.14	55	5.26 ± 1.30	34	5.16 ± 1.33	**0.002**	0.797	0.321
Falls in last year (No.)	159	0.54 ± 2.02	162	11.15 ± 50.13	52	3.04 ± 12.84	36	2.75 ± 3.86	**0.003**	0.572	0.125
Fear of falling ^§^	166	1.98 ± 1.31	159	3.20 ± 1.65	55	2.30 ± 1.68	35	3.01 ± 1.20	**<0.001**	0.266	0.196

^§^ Rating of fear of falling/quality of life from 1 (low) to 7 (high). SF12, short-form 12; PCS, physical component summary; MCS, mental component summary; PASE, physical activity scale for the elderly; CPF, composite physical function; BDI-II, Beck’s depression inventory II; ABC, activities-specific balance confidence scale. *p* < 0.05 are in bold.

## Data Availability

The data that support the findings of this study are available upon request from the corresponding author. The data are not publicly available due to the fact of privacy or ethical restrictions.

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
