# Peer review of "Parkinson’s Disease and Diabetes Mellitus: Individual and Combined Effects on Motor, Cognitive, and Psychosocial Functions"

_healthcare, 2023, doi:10.3390/healthcare11091316_

Round 1

Reviewer 1 Report

Dear Authors,

in the section "Statistics" you have declared that a two-way Analyses of Covariance (ANCOVA) was used. Please declare what is the covariate. 

Furthermore, please insert some graphs better to explain your results. 

Author Response

Response Healthcare Barter et al., paper

Dear Editors,

We are very excited to respond to the reviewers’ helpful critiques. We have carefully addressed each point and think that the manuscript is much improved. We look forward to hearing from you soon.

Sincerely,

Madeleine Hackney, Todd Prusin and authors.

Reviewer 1.

In the section "Statistics" you have declared that a two-way Analyses of Covariance (ANCOVA) was used. Please declare what is the covariate. 

RESPONSE: Thank you for the request to clarify. The covariates are sex, education and race.  In the section “Statistics”, our text now reads,
we controlled for these factors (sex, education and race) in comparisons of the four groups on continuous outcome variables using a stepwise linear regression with two-way Analyses of Covariance (ANCOVA). Four models were produced controlling for confounders: model 1: education, model 2: education and sex, model 3: education, sex, and race, and model 4: education, sex, race, and time with PD.”

Furthermore, please insert some graphs better to explain your results. 

RESPONSE: Thank you for the suggestion. We have included 3 graphs that show the important findings from all four groups (Figures 1, 2 and 3).

Reviewer 2:

The main aim of this study is to investigate if individuals with comorbid PD and DM experienced poorer functional ability compared to individuals with only one morbidity. The current study recruited 424 individuals (healthy participants, PD patients with and without DM). the enrolled a set of outcome measures to assess motor impairment, cognition, and psychosocial function. They reported some motor deterioration in patients with only PD and only DM.

Please cite the sentence “Most studies addressing the adverse impact of DM on disability progression in people with PD are not supported by the pathological confirmation of PD diagnosis.”

RESPONSE:

Thank you for pointing out this citation concern. We have decided to delete this sentence, as it was serving as a summary sentence for the following study finding presentations later in the paragraph. Removing this sentence does not detract from the message of the paragraph.

AND

“PD and DM can similarly negatively affect motor, cognitive, and psychosocial function.”

RESPONSE: Reference 6 now cites this sentence.

The authors stated in the inclusion criteria that the PD patients were tested either in the on or off-state. Indeed, the motor impairment may be aggravated at off state, thus, the author should address this in the limitation section.

RESPONSE: Thank you, the section “Limitations “states,
While differences between ON and OFF state were few (5/27 analyses were significant: Individuals ON Parkinsonian medication had faster backward gait speed, needed fewer steps and time to turn 360-degrees, had higher Corsi product scores, and made more correct subtractions per second during the TUG-cognitive compared to individuals who were tested while OFF their Parkinsonian medication (Supplementary Table 2), the limitation remains that individuals with PD that are OFF medication could be causing significant main effects of PD and the motor impairment may have been aggravated in an OFF state.”

The authors ignore the inclusion criteria to include the DM. In addition, PD patients in advanced stage have a cognitive impairment, the authors ignore the elimination of cognitive impermeant.

RESPONSE: All patients fulfilled the inclusion criteria as stated,
All eligible participants were aged 40-85 y, could walk at least 3 meters with or without assistance, had visual acuity of 20/40 or better in the best eye and were cognitively able to provide their own consent to the study.  In addition to these above criteria, the PD participants were also clinically diagnosed with PD by a board-certified neurologist with specialty training in movement disorders, exhibited 3 of the 4 cardinal signs of PD, exhibited clear symptomatic benefit from antiparkinsonian medications, and were in Hoehn & Yahr stages I-IV.”

Further, only 80% of people with PD develop cognitive impairment. Many people in stages 3 and 4 do not have a cognitive impairment preventing them from participating in the study and being able to consent on their own to the study. We included only these participants.

The authors should cite the outcome measures they used; they should cite all the outcome measures they used appropriately.

RESPONSE: Thank you, we have cited all the outcome measures, and used only valid, reliable and standardized measures, according to published methods.

The results section is reported well.

RESPONSE: Thank you for noting that the results section is reported well.

The first paragraph in the discussion should be cited appropriately.

RESPONSE: The first paragraph in the discussion provides a summary of the findings of this work. We have removed all sentences that derive from other sources therefore eliminating the need to cite anything in the first paragraph. We made some other changes to this paragraph to reflect the reviewers’ suggestions and believe it is much improved as a result.

Why the authors address this limitation “Significant differences between groups were observed for education, sex, and race.”

RESPONSE: In order to address this limitation, we used education, sex and race as covariates in the ANCOVA, as stated in the text below:

we controlled for these (sex, education and race) factors in comparisons of the four groups on continuous outcome variables using a stepwise linear regression with two-way Analyses of Covariance (ANCOVA). Four models were produced controlling for confounders: model 1: education, model 2: education and sex, model 3: education, sex, and race, and model 4: education, sex, race, and time with PD.”

By using these variables as covariates, we controlled for those confounders in our analyses.

Reviewer 3:

The study by Barter et al it is quite interesting, but its scientific novelty is low. Most of the deficits described by the authors are already known in PD and DM. Statistical interaction effects are few, and some of them are not associated with worsening motor or psychosocial conditions. The conclusion that comorbid DM-PD further impairs attention and slows organization/planning is not clearly supported by statistical data.

RESPONSE: Thank you for noting that the study is quite interesting. The reviewer is correct that the statistical interaction effects are few, which is the result of our very rigorous statistical analysis methods in determining an interaction effect. We agree with the reviewer that we needed to review our findings regarding organization / planning and have revised the manuscript as such. We also thank the reviewer for the frank questioning of the novelty of the work and have included a statement in the first paragraph of Discussion about why the work is important, which paraphrases the following statements. We believe our work is important and novel because: A) our findings corroborate previous findings, which has important scientific and clinical merit in its own right; B) we strengthen and highlight information that suggests that people with DM suffer from what is effectively, a movement disorder- given that those with DM had motor performance that was not unlike those with PD; and C) we present motor and cognitive performance on several standard, reliable and validated neuropsychological measures that were not previously presented in investigations concerning comparison of these populations.

Reviewer 2 Report

The main aim of this study is to investigate if individuals with comorbid PD and DM experienced poorer functional ability compared to individuals with only one morbidity.

The current study recruited 424 individuals (healthy participants, PD patients with and without DM). the enrolled a set of outcome measures to assess motor impairment, cognition, and psychosocial function. They reported some motor deterioration in patients with only PD and only DM.

Please cite the sentence “Most studies addressing the adverse impact of DM on disability progression in people with PD are not supported by the pathological confirmation of PD diagnosis.”

AND

“PD and DM can similarly negatively affect motor, cognitive, and psychosocial function.”

The authors stated in the inclusion criteria that the PD patients were tested either in the on or off-state. Indeed, the motor impairment may be aggravated at off state, thus, the author should address this in the limitation section.

The authors ignore the inclusion criteria to include the DM. In addition, PD patients in advanced stage have a cognitive impairment, the authors ignore the elimination of cognitive impermeant.

The authors should cite the outcome measures they used; they should cite all the outcome measures they used appropriately.

The results section is reported well.

The first paragraph in the discussion should be cited appropriately.

Why the authors address this limitation “Significant differences between groups were observed for education, sex, and race.”

Author Response

(The authors gave the same response as above.)

Reviewer 3 Report

The study by Barter et al it is quite interesting, but its scientific novelty is low. Most of the deficits described by the authors are already known in PD and DM. Statistical interaction effects are few, and some of them are not associated with worsening motor or psychosocial conditions. The conclusion that comorbid DM-PD further impairs attention and slows organization/planning is not clearly supported by statistical data.

Author Response

(The authors gave the same response as above.)

Round 2

Reviewer 3 Report

The authors have addressed all the issues, including new paragraphs and a good limitations chapter. The only question that remains is that they would set aside the conclusion that DM is a movement disorder. DM influences motor performance but is not a movement disorder.

Author Response

Comment: The authors have addressed all the issues, including new paragraphs and a good limitations chapter. The only question that remains is that they would set aside the conclusion that DM is a movement disorder. DM influences motor performance but is not a movement disorder.

RESPONSE: Thank you, we will change the language and will not suggest that DM is a movement disorder.